# Catholic Ownership, Physician Leadership and Operational Strategies: Evidence from German Hospitals

**DOI:** 10.3390/healthcare10122538

**Published:** 2022-12-14

**Authors:** Sandra Sülz, Ludwig Kuntz, Helena Sophie Müller, Michael Wittland

**Affiliations:** 1Erasmus School of Health Policy and Management, Erasmus University Rotterdam, Postbus 1738, 3000 DR Rotterdam, The Netherlands; 2Department of Business Administration and Health Care Management, University of Cologne, Dürener Str. 56-60, 50931 Cologne, Germany; 3Department for Nursing and Health Care, University of Applied Sciences and Arts Hannover, Blumhardt Str. 2, 30625 Hannover, Germany

**Keywords:** healthcare management, hospital management, physician leadership, chief medical officer, operational strategy

## Abstract

Previous research has revealed that Catholic hospitals are more likely follow a strategy of horizontal diversification and maximization of the number of patients treated, whereas Protestant hospitals follow a strategy of horizontal specialization and focus on vertical differentiation. However, there is no empirical evidence pertaining to this mechanism. We conduct an empirical study in a German setting and argue that physician leadership mediates the relationship between ownership and operational strategies. The study includes the construction of a model combining data from a survey and publicly available information derived from the annual quality reports of German hospitals. Our results show that Catholic hospitals opt for leadership structures that ensure operational strategies in line with their general values, i.e., operational strategies of maximizing volume throughout the overall hospital. They prefer part-time positions for chief medical officers, as chief medical officers are identified to foster strategies of maximizing the overall number of patients treated. Hospital owners should be aware that the implementation of part-time and full-time leadership roles can help to support their strategies. Thus, our results provide insights into the relationship between leadership structures at the top of an organization, on the one hand, and strategic choices, on the other.

## 1. Introduction

Catholic hospitals represent a large portion of hospitals in several countries. In the US, one out of nine hospitals is in the hands of a Catholic owner [1,2]; and in Germany, it is one out of seven hospitals [3,4]. Recently, it has been shown that religion, a main dimension of organizational culture, is an indicator for a specific “faithful” strategy. Concretely, Catholic hospitals follow a strategy of horizontal diversification and maximization of the number of patients treated, whereas Protestant hospitals follow a strategy of horizontal specialization [5]. Filistrucchi and Prüfer [5] trace the different strategies back to the theoretical foundations of the religion and thus show that religious ownership shapes operational strategies.

How do Catholic hospitals ensure that their values of communal bonds, traditions and altruism are embedded and embraced in their organization and strategy? This paper contributes by identifying and empirically testing a mechanism that facilitates these strategies. We argue that a specific aspect of hospital leadership makes a difference: Physician involvement in the hospitals’ top management team. Substantial and strategic changes in patient volume or horizontal specialization are subject to top management decision-making and leadership. Studies highlight a positive relationship between the degree of physician involvement and decision understanding, commitment, and quality [6,7]. Additional evidence exists that physicians in leadership in general use their power to strengthen specific quality-related issues. Rotar et al. [8], for instance, outline that hospitals whose physicians possess managerial roles and more formal decision-making responsibilities in areas pertaining to strategic hospital management are associated with higher implementation levels of quality management systems. Hospitals in systems led by a physician had higher quality ratings and more inpatient days per bed [9]. Furthermore, Kuntz and Scholtes [10] have shown that greater involvement of a hospital’s chief medical officer (CMO) in hospital leadership activities results in higher staff-to-patient ratios. With staff-to-patient ratios below target levels being related to higher mortality levels [11,12,13], increasing staff-to-patient ratios can be seen as a means to extend quality. Physicians in leadership thus have a peculiar role, as they are increasingly identified as being key for improving the organizational performance of hospitals [14].

Whether and how strategic decisions regarding patient volume and specialization levels can be influenced and steered by physicians in leadership is a relevant research question with implications for hospital owners. Owners decide on leadership structures, and therefore it is essential to know how the hospital leadership structure facilitates or impedes certain strategic decisions that need to be implemented and operationally executed [15,16]. Building on the work by Filistrucchi and Prüfer [5], and connecting it with physician leadership literature, the aim of this study is to identify and test a mechanism that connects ownership with operational strategies.

Our empirical study is based on data from 550 German hospitals. We combine a self- administrative survey concerning hospital leadership with publicly available information concerning the hospitals’ patient volume and service portfolio. Relying on simultaneous equation modelling, our empirical findings suggest that Catholic hospitals pursue their strategy via a different integration of the CMO into the top management team. In comparison to non-Catholic hospitals, Catholic hospitals are 9.5% points more likely to deploy a CMO in part-time position, resulting in different operational strategies. As such, the paper shows that the design of the CMO’s position within the leadership structure is a crucial factor for pursuing operational strategies. We conclude this paper by deriving suggestions for implementing physician leadership at the highest hospital hierarchy level.

## 2. Theory and Hypotheses

### 2.1. Hospital Ownership and Operational Strategies

Previous literature has shown that hospitals’ service portfolios differ across nonprofit hospitals owned by churches, other nonprofit hospitals, and for-profit hospitals [5,17,18]. Some of these differences are attributed to different value systems across hospital owners [5]. Catholic altruism is supposed to be community-focused, whereas Protestantism is more aligned with a stronger individual emphasis [19]. As a consequence, Catholic hospitals, in particular, have been identified as aiming to provide help for a broader range of patients, which results in higher patient volume and lower horizontal specialization compared to Protestant hospitals [5]. Protestant hospitals, on the other hand, are more focused on the individual, they aim to increase the benefit of an individual patient, and they seem to place more emphasis on productive efficiency, complex procedures, and technologies [5]. These differences in a hospital’s mission are reflected by different staff-to-patient ratios, with Protestant hospitals employing more doctors and specialists compared to Catholic hospitals [5].

In order to establish the link between hospital ownership and operational strategies, we therefore follow the argumentation by Filistrucchi and Prüfer [5] and hypothesize:

**Hypothesis 1.** 
*Hospitals’ operational strategies differ depending on the type of ownership, with Catholic hospitals fostering a strategy of higher patient volume and lower horizontal specialization.*


### 2.2. Physician Leadership and Operational Strategies

The seminal work about operations management by Skinner [20] and Wheelwright [21] has highlighted the crucial function of manufacturing as the “keeper of the corporate philosophy” [21]. Core processes and top executives in charge of them thus play a key role in defining guidelines for operational behavior and strategy. Empirical investigations that examine the role of Chief Supply Chain Officers (CSCOs) in firms, for example, point out that CSCOs, in particular, the leaders at the top level of the firm responsible for manufacturing, influence operational strategies. Particularly, strategies of diversification can be successful through the implementation of CSCOs in top management teams [22].

In hospitals, leading physicians are supposed to play a pronounced role, as they fulfil their leadership duties at the interface between management and medicine, giving them the main responsibility for hospital operations. In general, physicians—and CMOs in the hospital’s executive board as well—often perform their leadership position on a part-time basis, meaning that they still work as clinicians as well, usually as heads of clinical departments [14,23]. Maintaining a leadership position on a part-time basis is challenging, as managerial values often differ from professional values in medicine. Physicians aim to treat patients to increase medical outcomes without spending much consideration on financial and other organizational issues. General management and leadership duties, however, comprise strategic activities to achieve organizational or patient-related goals, even if these goals are inconsistent with the physicians’ personal or departmental goals. This may affect physician behavior in strategic decision making, especially if physicians fulfil leadership positions on a part-time basis, remaining closely linked to the physician profession and their physician peers. Therefore, physicians in leadership positions are forced to strike a balance between their clinical identity and their managerial duties. Competing logic, role ambiguity, and a shortage in time for fulfilling leadership tasks often decrease the physicians’ leadership abilities [23].

Focusing on the CMO, we argue that the ability of the hospitals’ CMO to determine operational strategies depends on the sources of power the CMO can use, which derives from the design of the CMO’s position as a full-time or part-time leader. Our arguments are built on applying theory from social psychology: Experimental studies show that the behavior of powerful actors depends on whether they feel independent or not. For example, Tost et al. [24] have shown that people in highly powerful positions act in ways to prevent themselves from losing their power, i.e., they act in ways that ensure their positions of power. We expect such mechanisms to influence operational strategies chosen by CMOs as well, depending on the base of power their leadership activities are built on.

Based on the theory of sources of power, we argue that the available sources of power for the CMO are defined by the design of the CMO’s position, consequently a part-time as opposed to a full-time position is exposed to different sources of power. General power theory differentiates between reward, coercive, expert, legitimate, referent, and informational forms of power [25,26,27]. In social interaction, individuals apply different forms of power to exert influence. Whereas the majority of the forms of power are basically founded in a leader’s personality, i.e., they emerge at an individual level. The ability to use legitimate power is determined by the design of the leadership role [28]. In contrast, expertise power depends on the ability and knowledge of the individual. In particular, it requires the recognition of these abilities by the corresponding peers whom the powerful individual aims to influence. Therefore, a full-time position created by the hospital owner relates to a higher level of legitimate power which may result in a more independent leadership role. This (and more time for leadership duties) may enable the full-time CMO to opt for more general changes to the service portfolio of the hospital, even if these changes conflict with the interests of the physician departmental leaders. Consequently, a full-time CMO might find it easier to pursue a specialization strategy that emphasizes certain services at the expense of others.

While a part-time CMO has legitimate power as well, the potential to use legitimate power to set operational strategies that reflect only the interests of a small subgroup of departmental leaders is far below that of a full-time CMO for two reasons. First, other forms of power such as expertise and reward power depend on the expertise and support of all physicians within the hospital. Second, part-time CMOs are often elected for a limited period (often two to four years) out of the entire group of physician departmental leaders. In this case, legitimate power is given not only by the organization or the hospital owner, but also by the other departmental leaders [28]. Thus, a part-time CMO needs to fulfil the interests of a broader group of departmental leaders and is consequently more likely to opt for operational strategies in line with the interests of the vast majority of physician peers. Therefore, a part-time CMO is more likely to push a strategy that increases patient volume across all specialties, as opposed to a full-time CMO who is able to act more independently, which may result in strategies characterized by higher horizontal specialization. Thus, we hypothesize:

**Hypothesis 2.** 
*A part-time CMO position is associated with an operational strategy characterized by higher patient volume and lower horizontal specialization.*


### 2.3. Hospital Ownership and Physician Leadership

Hospital owners decide on the structure of the top management teams of their organization. This decision is based on the values and the means to implement these values in the organization [29,30]. Specifically, in the context of hospitals, there is the option to decide whether a physician undertakes the CMO position in a full-time or a part-time capacity [10]. A CMO in a part-time position still treats patients in a specific hospital department and is fully responsible for their outcomes, whereas a full-time CMO exclusively works for the whole organization and has no responsibility for treatment plans of individual patients. Thus, a full-time CMO works fully outside of the traditional role of physicians and their medical associations. This might have consequences for the prevailing norms and values. Although research has shown that physicians in general are more altruistic than nonphysicians [31], individual values can change when physicians change to roles with more managerial tasks [32,33,34]. A management position comes with new demands and physicians can find themselves with a new role identity, which can be perceived as “leaving the professional family” [33]. Similarly, when blue-collar workers change to a role as a white-collar worker, the change is ongoing and associated with a change of values and behavior [35]. In the case of a Catholic hospital owner where the main values are altruism, tradition, and communal bounds [5,36], we argue that the CMO position is more likely to be designed as a part-time position, as it is more in line with these values. Furthermore, we build on empirical evidence for the connection between ownership and leading teams. Eldenburg et al. [29] investigated the board structures of different hospital ownership types in a sample of Californian hospitals. They found that board characteristics often have organization and owner type-specific constraints and history. Specifically, hospitals in governmental ownership were more likely to include politicians on the board [30]. We therefore hypothesize that hospitals in Catholic ownership prefer a part-time CMO. Together with the preceding hypothesis that a part-time CMO position is associated with an operational strategy characterized by higher patient volume and a lower specialization, this leads us to our mediation hypothesis:

**Hypothesis 3.** 
*The effect of the type of ownership on hospital operational strategy is mediated by the design of the CMO position, with Catholic owners preferring a part-time CMO.*


## 3. Materials and Methods

### 3.1. Setting

We used the German hospital setting to test the hypotheses. This setting had the advantage of exhibiting considerable ownership variety. German hospitals are either public or private organizations, with public hospitals owned by cities, districts, counties, or the state, and private hospitals owned by entities with either a for-profit or a nonprofit orientation. The owners of nonprofit hospitals frequently have religious affiliations with Catholic and Protestant hospitals, accounting for the majority of nonprofit hospitals [5].

As described by Schmidt-Rettig [37], German hospitals are characterized by a fairly homogeneous organizational structure. German hospitals typically consist of multiple clinical departments organized by medical specialties. Each clinical department is led by a senior physician accountable for the department’s financial budget and clinical operations. Cure and care departments are supported by departments responsible for the hospital’s administration and general business operations. At the top of the hospital’s hierarchy is the top management team. The top management team usually consists of a commercial director, a nursing director, and a CMO, and can be extended by further physicians or a single CEO with final decision-making authority. The commercial director, the nursing director, and the CMO have well-defined and well-regulated roles and responsibilities, covering the overall hospital-level administrative and managerial functions. While the commercial director usually has a full-time employment to monitor the hospital’s financial operations and performance, the nursing director acts as the representative of the hospital’s nursing staff and advocates their concerns. The CMO represents the hospital’s physicians and is responsible for the hospital’s clinical operations. By law, only physicians can be appointed as CMOs. German hospitals can employ a part-time CMO, who is usually hired internally from among the senior physicians leading the clinical departments. The part-time CMO therefore has to accommodate the role with the responsibility of practicing medicine and running a department. Alternatively, hospital governing boards can appoint a full-time CMO. Although it is highly possible that these physicians have gained experience as heads of a clinical department, they spend 100% of their time in the CMO role and are not accountable for a specific department. Full-time CMOs are typically employed on management contracts, instead of practicing as clinicians. These full-time positions are promoted broadly, and vacancies are filled by either internal or external candidates.

### 3.2. Data and Sample

For the purpose of this study to analyze the impact of ownership on operational strategies, we used survey data and publicly available information. The survey was not set up for the specific purpose of this study, but it contained information relevant for this research objective.

Information about the CMO position was obtained through using data from a self- administrated survey conducted in 2008. The survey had the general aim to assess the role of the CMO within the leadership team (anonymized). The mail survey was executed in 2008 and identified 1913 target hospitals. Each executive office of the target hospitals received a cover letter and single-page questionnaire. Besides stating the general aim of the survey, the cover letter outlined that respondents could participate in the research by means of the questionnaire, i.e., consent was provided through returning the questionnaire. The cover letter stressed that survey responses should capture the dominant situation over the past ten years. In particular, we were interested in whether the CMO was in a full-time or in a part-time position. The cover letter also pointed out that the executive director with the best knowledge of the prior ten years should respond to the survey, and these persons did not necessarily have to be the CMOs themselves. Another director may well have better knowledge of the prevalent situation over the past ten years than the incumbent CMO, especially in the case of part-time CMOs. Much of a part-time CMO’s time is spent in joint committee meetings, and the degree of this participation can be assessed by other directors. In total, we received 675 returned questionnaires. Responses with significant missing data and obvious inconsistencies were discarded, yielding a total data sample of 604 hospitals, which equates to a response rate of 31.6%. This is a reasonable response rate if we take into account that our survey targeted executives at the top of an organization [38].

We complemented the survey with publicly available information derived from the annual quality reports of German hospitals for the year 2008. These reports are freely available online [39] and contain structural information at the overall hospital level, i.e., ownership type (private for-profit, private nonprofit, public), teaching status, number of registered beds. For all nonprofit hospitals, we also checked whether the hospital had a religious affiliation. The religious affiliations were identified with additional information from the hospital’s homepage if necessary.

Besides structural information, the annual quality reports also provided a comprehensive overview of the hospital’s patient volume and service portfolio by listing, for each clinical department, and at least the ten diagnosis codes with the highest patient volume. Diagnosis codes were provided in line with the international classification of diseases (ICD) scheme at granular three-digit levels (e.g., I48 atrial fibrillation and flutter; C50 malignant neoplasm of the breast). Within the ICD scheme, the first digit indicates the ICD chapter and refers to organ systems (I: diseases of the circulatory system) or disease categories (C: neoplasm). We relied on the ICD chapter information to identify services for our operational strategies.

Finally, we used data from the Federal Office for Building and Regional Planning [40], which we matched to our hospital data based on the hospitals’ ZIP codes. This database allowed us to control for the historic cure and care demand within the hospitals’ catchment areas in 2008. To create our final sample, we merged the different databases. We excluded hospitals whose ownership status could not be unambiguously attributed to one of the generic types (private for-profit, private nonprofit, public) (N = 4 hospitals). After matching the different databases, our sample composed of N = 550 individual hospitals.

### 3.3. Variables

#### 3.3.1. Operational Strategies at a Hospital Level

Our operational strategies were derived from the annual quality reports. For each hospital $h^{\;}$, the volume strategy $V_{h}\;$ was operationalized as the ln-transformed total patient volume admitted to hospital $h^{\;}$. Patient volume is a commonly used as an indicator for hospital treatment activity [41].

The second measurement related to the hospital’s level of horizontal specialization. A hospital can be active in different services, as indicated by a patient’s primary diagnosis, categorized in different ICD chapters [5]. The fewer different services there are in which a hospital is active, the more horizontally specialized the hospital’s service portfolio is [5]. We took this thought one step further and not only considered the number of different services, but also whether some individual services accounted for disproportionately more patients, i.e., whether they receive a higher focus relative to other services. With ${\tilde{V}}_{h}$ denoting the hospital’s annual patient volume and $A_{ih}\;$ denoting the hospital’s annual patient volume in service $i_{\;}$, the proportion $A_{ih}\;/\;{\tilde{V}}_{h}$ indicated the hospital’s emphasis on service $i_{\;}$ (for an equivalent operationalization, see the recent OM studies in [42,43,44,45]). To determine the hospital’s level of horizontal specialization, we relied on the Herfindahl–Hirschman Index (HHI), which has been previously used to measure a variety of tasks [46,47]. For each hospital $h^{\;}$, we calculated $HHI_{h}=\sum_{i}{\left(A_{i}h\;/\;{\tilde{V}}_{h}\right)}^{2}$. At its maximum, the HHI took the value 1, indicating a fully specialized service portfolio in which the hospital’s patient volume belonged to only one service. In the case of the hospital having an equal patient share in each service, the HHI took the minimum possible value of 1/N, with N denoting the number of different services the hospital provides.

#### 3.3.2. Hospital Ownership and Physician Leadership

We used the following two dichotomous measures for hospital ownership and physician leadership: $Cat_{h}$ was a dichotomous variable equal to 1 if the owner of hospital h was a Catholic nonprofit organization, and 0 indicated otherwise. To assess physician leadership as a part-time CMO, $Ρ{Τ}_{h}$, a dichotomous variable was equal to 1 if the role was part-time in the hospital h, and 0 if otherwise.

#### 3.3.3. Control Variables

Several factors might have been related to the operational strategies and the hospital ownership (physician leadership) and could therefore have confounded our results: the vector of hospital-level controls $H_{h}$ contained the number of registered beds as a proxy for hospital scale, whether the hospital operated at multiple locations (=1; 0 otherwise), and whether the hospital had a teaching affiliation (=1; 0 otherwise).

Similar to Filistrucchi and Prüfer [5], we captured market factors via the hospital’s distance $Dis_{h}$ to the nearest neighboring hospital using ZIP geocodes and the geocode command in Stata [48]. In addition, the vector $Dem_{h}$ captured historic cure and care demand factors within the hospital’s close catchment area. The hospital’s close catchment area was approximated via the district in which the hospital was located. For each district, we used data from BBR [40] and determined the average income per inhabitant and the percentage of senior inhabitants aged 75 years or older.

### 3.4. Statistical Analysis

Our two dependent variables were theoretically and empirically related and therefore should not have been analyzed separately. We therefore tested the mediation effect by means of two simultaneous equation models. Model (1) looks as follows:$$V_{h}=\beta_{0}^{V}+\beta_{Cat}^{V}Cat_{h}+\beta_{H}^{V}H_{h}+\beta_{Dis}^{V}Dis_{h}+\beta_{Dem}^{V}Dem_{h}+\epsilon_{h}^{V}$$
$$HHI_{h}=\beta_{0}^{H}+\beta_{Cat}^{H}Cat_{h}+\beta_{H}^{H}H_{h}+\beta_{Dis}^{H}Dis_{h}+\beta_{Dem}^{H}Dem_{h}+\epsilon_{h}^{H}$$
where the errors $\left(\epsilon_{h}^{V},\epsilon_{h}^{H}\right)$ were sampled from a multivariate standard normal distribution
ϵHV ϵhH~Ν00,σv2 ρVH ρVH σH2
and $\rho_{VH}\;$ is the correlation between unobservable factors affecting both operational strategies.

In Model (2), we integrated a separate equation for the part-time CMO position $Ρ{Τ}_{h}$. We assumed that the part-time CMO position was a function of the hospital ownership, the hospital structure, and the competitive situation, as captured through the difference to the next provider:$$V_{h}=\gamma_{0}^{V}+\gamma_{Cat}^{V}Cat_{h}+\gamma_{ΡΤ}^{V}Ρ{Τ}_{h}+\gamma_{H}^{V}H_{h}+\gamma_{Dis}^{V}Dis_{h}+\gamma_{Dem}^{V}Dem_{h}+\epsilon_{h}^{V}$$
$$HHI_{h}=\gamma_{0}^{H}+\gamma_{Cat}^{H}Cat_{h}+\gamma_{ΡΤ}^{V}Ρ{Τ}_{h}+\gamma_{H}^{H}H_{h}+\gamma_{Dis}^{H}Dis_{h}+\gamma_{Dem}^{H}Dem_{h}+\epsilon_{h}^{H}$$
$$Ρ{Τ}_{h}^{*}=\gamma_{0}^{Ρ}+\gamma_{Cat}^{Ρ}Cat_{h}+\gamma_{H}^{Ρ}H_{h}+\gamma_{Dis}^{Ρ}Dis_{h}+\gamma_{Dem}^{Ρ}Dem_{h}+\epsilon_{h}^{Ρ}$$
$$Ρ{Τ}_{h}=1[Ρ{Τ}_{h}^{*}>0]\;$$
$$\left(\begin{array}{c} \epsilon_{h}^{V}\\ \epsilon_{h}^{H}\\ \epsilon_{h}^{Ρ} \end{array}\right)\sim Ν\left[\left(\begin{array}{c} 0\\ 0\\ 0 \end{array}\right),\left(\begin{array}{c} \sigma_{v}^{2}\\ \rho_{VH\;}\\ \rho_{VΡ\;}\; \end{array}\begin{array}{c} \rho_{VH}\\ \sigma_{H}^{2}\\ \rho_{HΡ\;} \end{array}\;\begin{array}{c} \rho_{VΡ}\\ \rho_{hΡ}\\ 1 \end{array}\right)\right],$$
whereby $\rho_{VP\;}\left(\rho_{HP}\right)$ captures the extent to which unobservable factors affect the volume strategy (horizontal specialization strategy) and the decision to deploy a part-time CMO simultaneously. If there were unobserved factors that made it more likely (or less likely) for a hospital to pursue a volume strategy and deploy a part-time CMO, the error terms $\epsilon_{h}^{V}$ and $\epsilon_{h}^{P}$ would have been positively (or negatively) related, resulting in $\rho_{VP\;}$ > 0 ($\rho_{VP\;}$< 0). Both models were estimated in Stata 16.1 with the user-written STATA command cmp [49].

To test for mediation, we compared the coefficients estimated in Models (1) and (2). The coefficients $\beta_{Cat}^{V}\;$ and $\beta_{Cat}^{H}$ estimated in Model (1) represent the *full effect*, i.e., the direct effect plus the indirect effect transmitted through the CMO position, of Catholic hospitals on operational strategies. The coefficients $\gamma_{Cat}^{V}$ and $\gamma_{Cat}^{H}$ estimated in Model (2) represent the *direct effect* of Catholic ownership on operational strategies. If $\gamma_{Cat}^{V}$ < $\beta_{Cat}^{V}$ and $\gamma_{Cat}^{H}$ < $\beta_{Cat}^{H}$, then the effect of Catholic ownership is mediated by the CMO position. We tested the difference in coefficients using a bootstrapping procedure. First, we drew a random sample of hospitals with replacement. Then, we estimated Models (1) and (2) and computed the differences in the coefficient, i.e., $\Delta_{V}=\gamma_{Cat}^{V}-\beta_{Cat}^{V}$ and $\Delta_{H}=\gamma_{Cat}^{H}-\beta_{Cat}^{H}$. We repeated this for 1000 bootstrap samples and determined the 95% CI for $\Delta_{V}$ by means of its 2.5th and 97.5th percentile (likewise for $\Delta_{H}$).

## 4. Results

Table 1 provides mean standard deviation and correlations of our variables. The descriptive statistics were in line with our hypotheses, i.e., we see that Catholic ownership correlates with higher volume, lower horizontal specialization, and higher chances of part-time CMO positions. The data also showed that a part-time CMO position correlated with higher levels of patient volume and lower levels of horizontal specialization. Finally, the data indicated a strong negative correlation between patient volume and specialization levels, which supported our choice of a simultaneous equation model.

We report the model results in Table 2, which lists the coefficient estimates in the upper panel and the estimated correlation coefficients of the error terms in the second panel. In both models, we found a strong correlation between the operational strategies ($\rho_{VH\;}$= −0.573, −0.556, respectively), which supported our choice for a simultaneous model.

Hypothesis 1, which states that Catholic hospitals foster a strategy of higher patient volume and lower horizontal specialization, is supported with the full effect estimated in Model (1) ($\beta_{Cat}^{V}$= 0.414, *p* < 0.001; $\beta_{Cat}^{H}$= −0.179, *p* < 0.001).

Hypothesis 2 posits that a part-time CMO position is associated with an operational strategy characterized by higher patient volume and lower horizontal specialization. Model (2) also support this hypothesis ($\gamma_{PT}^{V}\;$= 0.594, *p* < 0.001; $\gamma_{PT}^{H}\;$= −0.194, *p* < 0.05). Finally, Hypothesis 3 stipulates that the effect of the type of ownership on hospital operational strategy is mediated by the design of the CMO position, with Catholic owners preferring a part-time CMO. We also found support for this hypothesis. Catholic hospitals are indeed associated with a higher probability of deploying a part-time CMO ($\gamma_{\mathrm{Cat}\;}^{P}$ = 0.509, *p* < 0.05) and including the variable part-time CMO position in Model (2) mitigated the explanatory strength of the variable $Cat_{h\;}$. A reduction in the explanatory strength can be quantified via the difference in coefficients $\Delta_{V}=\gamma_{Cat}^{V}-\beta_{Cat}^{V}$ and $\Delta_{H}=\gamma_{Cat}^{H}-\beta_{Cat}^{H}$. Figure 1 plots these differences obtained from bootstrapping. While there was considerable variance, the overwhelming majority of the bootstrap results indicated that these differences are different from 0 (95% CI $\Delta_{V}$ [−0.141; −0.021], $\Delta_{H}$ [0.003; 0.067]). Taken together, our findings suggest that the effect of Catholic ownership on operational strategies is partially mediated by the CMO position.

Table 3 outlines the predictions and the average partial effects derived from Model (2). The predictions in the first panel were obtained by calculating the estimated operational strategy and leadership structure for each hospital in the counterfactual scenarios that the owners of all hospitals were Catholic or non-Catholic. We obtained the untransformed patient volume as follows: Patient volume = $\exp\left({\hat{V}}_{h}\right)\times \exp\left({\hat{\sigma }}_{V}^{2}/2\right)$ [50]. Shifting from a Catholic to a non-Catholic owner was linked to a 3917 reduction in patient volume, which represented a reduction of 31.9% relative to the mean patient volume in the sample. Likewise, the HHI of Catholic hospitals was, on average, 0.161 points lower compared to non-Catholic hospitals, which is a difference of 64.4% relative to the mean HHI in the sample. Finally, the probability of deploying a part-time CMO was 9.5 percentage points higher for Catholic hospitals compared to non-Catholic hospitals.

The predictions in the second panel were obtained by calculating the estimated operational strategy for each hospital in the counterfactual scenarios that the CMO positions of all hospitals were part-time or full-time. Shifting from a part-time to a full-time CMO position is linked to a reduction in patient volume by 4623, which represents a reduction of 37.6% relative to the mean patient volume in the sample. Likewise, the HHI of hospitals with a part-time CMO was, on average, 0.194 points lower compared to hospitals relying on full-time CMOs, which is a difference of 77.6% relative to the mean HHI in the sample. This shows that all effect sizes were also practically relevant.

## 5. Discussion

### 5.1. Main Findings

Our results show that a full-time or a part-time position of a CMO fosters different operational strategies. Hospitals with a part-time CMO are more likely to implement a strategy of increasing patient volume and lower horizontal specialization. Using a power perspective, we argue that the influence of a part-time CMO is mainly based on expertise power, whereas the influence of a full-time CMO is rather based on legitimate power and therefore more independent from other physicians in the hospital. Thus, a connection exists between a part-time CMO and the other physicians within the organization making it more difficult to reallocate resources or even to completely cancel services. We also show that Catholic hospitals rely more often on a part-time CMO and use this tension as a mechanism to foster their strategies. This supports our explanation of the recently identified differences in strategy patterns compared to hospitals in non-Catholic ownership [5]. Our results also add to previous findings from experimental research by Brinol et al., Galinsky et al., and Tost et al. [24,51,52], underlining the consequences of feeling powerful independently of others for implementing operational strategies.

Previous literature in the general business administration context has already indicated that non-CEO roles in top management teams can influence the overall performance of an organization [22,53]. In the healthcare context, physicians in executive roles matter [8,9]. We complement this literature by showing that part-time or full-time role designs influence performance. By designing a position such as a full-time role, hospital owners establish a position equipped with a high level of legitimate power, which enables the CMO to implement a comprehensive strategy for the whole hospital based on key areas. A part-time design, however, makes this more difficult, as power may be basically derived from expertise power that depends on the recognition of physician peers.

### 5.2. Implications for Research and Practice

Volume and horizontal specialization have been identified as a lever to improve cost efficiency and medical outcomes in hospitals [42,43,44,54]. Thus, being able to influence operational strategies is important in improving overall performance in many dimensions. Hospital owners may influence operational performance by designing executive committees [55]. First, to realize a comprehensive strategic approach, addressing a service portfolio characterized by a high level of specialization, our results suggest appointing a CMO in a full-time position. Second, if the hospital’s objective is to increase patient volume substantially over all departments—as is the case with Catholic hospitals—our findings suggest implementing a part-time CMO position. Taken together, our results show that the design of single positions in executive committees can help to implement strategies. Concretely, we advise owners of organizations to think about the part-time members in coherence with their strategies, as they may remain connected with their professional peers within the organization.

As we have not addressed questions of personality due to the design of our empirical survey, we are not able to address further aspects besides the design of the CMO position. Nevertheless, the CMO personality may also influence operational strategies. Following the upper echelon approach introduced by Hambrick and Mason [56], organizations can be understood as reflections of their top executives [56,57], placing additional emphasis on the personality of individuals in leadership roles. Strategic choices—decisions at the top of an organization that are characterized by a high level of complexity and significance [58]—are based on the personalities of top executives. While the executives’ knowledge and cognitive abilities determine possible courses of action, their individual values and goals are an important basis for the evaluation of these alternatives [56,57,59,60,61]. Future research should therefore consider the personality of top management team members, leadership styles, and the cooperation between these members to increase our understanding further on how leadership influences operational strategies.

### 5.3. Limitations

The data that we used for testing the hypotheses were not perfect. The initial survey was not set up for the specific purpose of this study and it was conducted in 2008. However, the survey contained information whose reuse was considered relevant for complementing Filistrucchi and Prüfer’s novel research insights. Filistrucchi and Prüfer use hospital data from 2006 and 2008, which overlaps with this study’s time frame [5]. This allowed us to directly link our findings to the work of Filistrucchi and Prüfer because we considered a similar period and face comparable regulatory, technological, and organizational circumstances [5]. We also acknowledge that the sample was not fully representative of the German hospital landscape. The sample was composed of 17% private for-profit, 38% public, and 45% nonprofit hospitals, which had an average of 372 beds and serve 12,299 annual inpatient cases. In 2008, Germany had 31% private for profit, 32% public, and 37% nonprofit hospitals with an average of 242 beds and 8411 annual inpatient cases [4]. The sample was thus slightly biased towards larger nonprofit hospitals. In this sample, private for-profit hospitals had an 84% likelihood of part-time CMO appointments, which was in the range of the other non-Catholic hospitals (75–86%). At the same time, private for-profit hospitals tended to have lower annual patient volumes (8420 inpatient cases on average) and higher HHI scores. With private hospitals thus being underrepresented in this sample, the group of non-Catholic hospitals in our sample therefore had a higher patient volume and lower concentration score than what was expected in the national average. The difference in volume and horizontal diversification strategy between Catholic and non-Catholic hospitals thus becomes smaller, which makes it even more challenging to identify differences in strategies. Consequently, the results more likely underestimate the difference, rather than overestimate it.

With respect to the econometric analysis and the empirical results one might argue that part-time CMOs might be more likely positioned in larger hospitals and are therefore associated with a higher patient volume. However, this concern was mitigated since we controlled for hospital scale via the number of registered beds and whether the hospital had multiple facilities at different locations.

Furthermore, one might be concerned with the unobserved heterogeneity affecting the decision to deploy a part-time CMO and the operational strategy simultaneously. However, this was explicitly accounted for in the simultaneous equation model. Unobservable heterogeneity affecting the volume strategy (specialization strategy) and the decision to deploy a part-time CMO was captured through the correlation coefficient $\rho_{VP}\;\left(\rho_{HP}\right)$. Neither estimate showed evidence of a strong correlation $\rho_{VP}$ = −0.093, *p* > 0.05 ($\rho_{HP}$ = 0.103, *p* > 0.05), i.e., empirically there was no indication of unobserved heterogeneity affecting the CMO position and the operational strategies.

To further address alternative explanations and potential issues of reversed causality and to validate our theoretical argumentation, we conducted an ex post survey with top managers, board members, and hospital experts. This was particularly important given that one might argue that the CMO position is a consequence of the hospital’s patient volume and service portfolio instead of the hypothesized antecedent of operational strategies. To address this concern, we invited hospital CEOs, members of hospital supervisory boards, and hospital consultants to participate in a survey. The survey was designed to assess the rationale for the CMO position, the expected consequences of the position, and the differences between full-time and part-time CMOs. We purposefully selected the respondents based on their function and experience because we wanted to have an expert panel with profound knowledge of the German hospital setting. We conducted the survey in 2021, approached respondents, asked them to further distribute the survey within their network, and received N = 18 responses. Through returning the questionnaire, the respondents provided their consent for study participation. All members of the panel indicated that hospital owners make a conscious decision whether to design the CMO position as a full-time or a part-time job, with expected consequences for (amongst other aspects) hospital strategy, development of centers of expertise, and networking with professional associations. This gives credibility to our claim that the CMO position is likely to affect the hospital volume and service portfolio. The respondents further indicated that full-time and part-time CMOs differ fundamentally with respect to personal goals, which indicates not only that the design of the position, but also that the personality of the CMO, could play a role in influencing operational strategies—a point that we have discussed before.

## 6. Conclusions

Building on the work by Filistrucchi and Prüfer [5], who outline why Catholic hospitals pursue a particular strategy, we identified and empirically tested a mechanism that facilitated these strategies. Specifically, we showed that “Faithful strategies” pursued by Catholic hospitals are facilitated by physician leadership structures relying on a part-time CMO positions. Our findings complement the body of literature showing that physician leadership has implications for performance. With the current trend to professionalize the role of the CMO and an increasing awareness for physician leadership, our findings are also of interest to hospital boards who are adjusting their structure. Given that the design of single positions in executive committees can help to implement strategies, owners of organizations are advised to think about the part-time members in coherence with their strategies, as these part-time members may remain connected with their professional peers within the organization.

## Figures and Tables

**Figure 1 healthcare-10-02538-f001:**
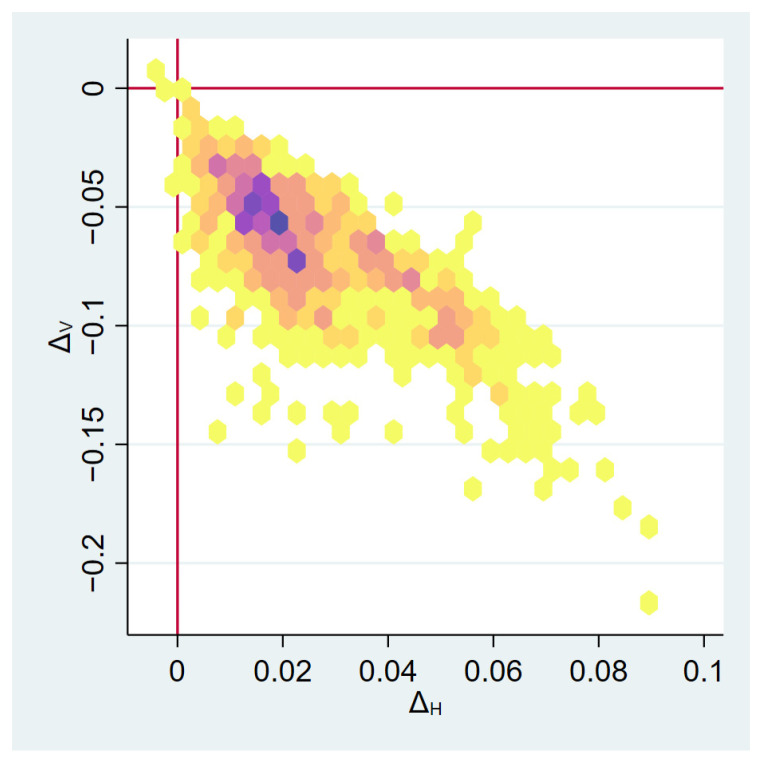
Differences in coefficients derived from 1000 bootstrap samples.

**Table 1 healthcare-10-02538-t001:** Descriptive statistics and correlations.

			Correlations			
	Mean	SD	$V_{h}$	$HHI_{h}$	$Cat_{h}$	$Ρ{Τ}_{h}$
Log. Patient volume ($V_{h}$)	8.920	1.098				
$HHI_{h}$	0.250	0.282	−0.5996 *			
Catholic ownership $Cat_{h}$	21.5%		0.0933 *	−0.2015 *		
Part-time CMO ${PT}_{h}$	85.5%		0.1629 *	−0.2209 *	0.1151 *	
Total number of beds	372.296	388.005	0.6699 *	−0.2567 *	−0.062	−0.0235
Hospitals with multiple locations	1.8%		0.1562 *	−0.0739	−0.0048	−0.0597
Teaching hospital	42.9%		0.4545 *	−0.1853 *	−0.0325	0.0347
Distance to next hospital	4.231	6.324	−0.0821	−0.1389 *	−0.1448 *	0.048
Average income per inhabitant	1576.984	244.460	−0.0735	0.1129 *	−0.0382	−0.1208 *
Percentage of senior inhabitants	8.8%		−0.0055	−0.0091	−0.1478 *	−0.0534

N = 550, * *p* < 0.05.

**Table 2 healthcare-10-02538-t002:** The association of Catholic ownership and part-time CMO positions with operational strategies.

	Model (1)		Model (2)		
Variables	$V_{h}$	$HHI_{h}$	$V_{h}$	$HHI_{h}$	$Ρ{Τ}_{h}$
Catholic	0.414 ***	−0.179 ***	0.358 ***	−0.161 ***	0.509 *
	(0.060)	(0.020)	(0.061)	(0.021)	(0.204)
PT			0.594 ***	−0.194 *	
			(0.177)	(0.086)	
Beds	0.002 ***	−0.000 ***	0.002 ***	−0.000 ***	−0.000
	(0.000)	(0.000)	(0.000)	(0.000)	(0.000)
Locations	−0.473	0.083	−0.357	0.046	−0.707
	(0.379)	(0.058)	(0.351)	(0.055)	(0.416)
Teaching	0.529 ***	−0.070 **	0.510 ***	−0.063 **	0.170
	(0.088)	(0.025)	(0.082)	(0.024)	(0.156)
Distance	0.011 *	−0.010 ***	0.009 *	−0.010 ***	0.015
	(0.005)	(0.002)	(0.004)	(0.001)	(0.012)
Income	−0.000	0.000	0.000	0.000	−0.001 *
	(0.000)	(0.000)	(0.000)	(0.000)	(0.000)
Seniors	0.052	−0.013	0.065*	−0.018	−0.094
	(0.032)	(0.011)	(0.032)	(0.010)	(0.071)
Constant	7.600 ***	0.464 **	6.834 ***	0.715 ***	2.747 **
	(0.431)	(0.144)	(0.498)	(0.165)	(0.865)
Error correlations					
$\rho_{VH},\;\rho_{VP}$		−0.573 ***		−0.556 ***	−0.093
		(0.038)		(0.038)	(0.079)
$\rho_{HP}$					0.103
					(0.158)

N = 550, Robust standard errors in parentheses. *** *p* < 0.001, ** *p* < 0.01, * *p* < 0.05.

**Table 3 healthcare-10-02538-t003:** Predictions including 95% CI and average partial effects derived from Model (2).

	Hospital Ownership		
	Catholic	Non-Catholic	APE
Patient volume	13,017 [11,909; 14,229]	9100 [8434; 9819]	3917
HHI	0.124 [0.097; 0.152]	0.285 [0.258; 0.311]	−0.161
Probability Part-time CMO	92.9% [88.0%; 97.7%]	83.4% [79.9%; 86.9%]	9.5 pp
	CMO Position		
	Part-time	Full-time	APE
Patient volume	10,711 [10,037; 11,413]	6088 [4218; 8169]	4623
HHI	0.222 [0.193; 0.251]	0.416 [0.267; 0.565]	−0.194

pp: percentage points. APE: Average partial effect.

## Data Availability

Publicly available datasets were analyzed in this study. This data can be found here: https://g-ba-qualitaetsberichte.de/#/search (accessed on 15 May 2021). The survey data presented in this study are available on request from the corresponding author.

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
