# Peer review of "Catholic Ownership, Physician Leadership and Operational Strategies: Evidence from German Hospitals"

_healthcare, 2022, doi:10.3390/healthcare10122538_

Round 1
Reviewer 1 Report
The manuscript presents a search for an interesting connection between hospital management and religious doctrine.
The summary is too general and does not contain any specific conclusions from the study.
In the introduction - there is no reference to religious doctrine and linking it to doctors' leadership roles and higher quality indicators.
I don't see a description of the relationship between Catholic altruism and Protestant individualism. What is the scientific evidence for this? (from verse 80)
Although research has shown that physicians in general are more altruistic than non-physicians, individual values can change when physicians change to roles with more managerial tasks - can you explain?
You sholud give more information about reserach - questionaire.
Data for 2008 quite outdated. Isn't there more recent data available?
Conclusions - they are written as conclusions from the research already done, not what the authors have done - are for improvement.
Author Response
Thank you very much for your helpful feedback. Please find our response attached.

Reviewer 2 Report
The paper deals with an extremely important topic for hospital management and leadership, for health policies and for German, European and global health systems. It makes an important contribution to the scientific evidence.
The topic Catholic Ownership, Physician Leadership & Operational Strategies: Evidence from German Hospitals is of great importance and interest and is very well described and justified.
The introduction is very focused on the topic and up to date, both scientifically and in relation to the issue.
The aim of this study should be clearly identified: The aim of this study is ... instead of appearing in line 62: (...) "this paper contributes by"(...)
Considering the aim of this study as identifying and testing the mechanism that connects ownership with operational strategies, the authors could have defined two objectives in order to match the delineation of the totality of the study. Because the study had two phases and this is not properly described in the abstract, in 3. Materials and Methods and even in 3.2 Data and Sample.
Chapter 2. Theory and Hypotheses - the construction of the study hypotheses is very well justified and with reference to the theoretical construct. They could refer to leadership theories as this is also a study on leadership.
Chapter 3. Materials and Methods, the research question that led to the study design is missing and the characterization of the type of study is missing.
Chapter 3.1. Setting very well described and related to the theme of the study.
Chapter 4. Results very well described.
The most fragile issue of this paper is the justification of scientific timeliness, considering data collection in 2008. After thirteen years since this data collection, an additional study was conducted in 2021, indicated in chapter 5.3. Limitations and further research. The authors should justify very well, this gap of thirteen years of the study. On the other hand, in my humble opinion, it does not make sense to present the additional study in chapter 5.3. Limitations and further research when they make no reference in 3. Materials and Methods. Why did they not consider the two-phase study in 3. Materials and Methods? Does the additional study only appear in chapter 5?
Where is the reference to the ethical issues of the study? If it is a scientific study using data collection through a questionnaire (lines 473 to 489)...
Chapter 5.3. the limitations of the study should be separate from the further research.
In the limitations of the study, the fact that the data collection took place thirteen years ago should be indicated and well justified.
The conclusions are too summary. They should be more developed and with reference to the main findings of the study, to reinforce all the interest that the paper has.
In the conclusions reference should be made to the articulation and complementarity of the two phases of the study as well as in the abstract itself.
Congratulations for the study!
Author Response
Thank you for your constructive feedback and the appreciation of the research topic. Please find our responses attached.

Reviewer 3 Report
The manuscript conducted research regarding whether physician leadership mediates the relationship between ownership and operational strategies. For that reason, the authors conducted an empirical study in German settings.
The manuscript focuses on an interesting topic, is well written, and includes adequate citations throughout the document. However, the manuscript is based mainly on data from a survey conducted in 2008 and annual quality reports of German hospitals for the year 2008.
In this regard, the data comes from sources obtained more than 14 years ago. This represents an important weakness of the manuscript, especially since several changes have occurred in healthcare settings, including technological and organizational changes, thus reducing the possibility to present updated and useful information for the current context. Particularly, readers might be interested in actual context including the Covid-19 Pandemic effects.
Related to this issue, the authors stated, “To validate our theoretical argumentation, and to address alternative explanations and potential issues of reversed causality, we conducted a survey with top managers, board members, and hospital experts.”
To do so, the authors added “We conducted the survey in 2021, approached respondents, asked them to further distribute the survey within their network, and received N=18 responses.” In this manner, how can such sample serve to validate the theoretical argumentation?
Moreover, only 45% of the respondents were from a non-profit hospital (the ones having affiliation with Catholic and Protestant hospitals) for which the hypotheses were tested, thus reducing even more the validation of the theoretical argumentation.
Based on these observations, I do not recommend this manuscript for publication.
Author Response
Thank you for your valuable suggestions on this paper. Please find our response attached.

Reviewer 4 Report
Dear authors
Thank you so much for submitting your paper "Catholic Ownership, Physician Leadership & Operational Strategies: Evidence from German Hospitals" to this esteemed journal 'Healthcare (ISSN 2227-9032)'. I read your paper and gave my concern down here for improving it further:
1. The motivation of the paper is not properly justified. I mean that the problem statement of the study. The author might bring out the problem in the first paragraph.
2. The gap of the study can be listed down in the last paragraph of the study. If the gap in the literature or methodology or theoretical insights is not signified anywhere, the originality of the study might not be believed.
3. The authors wrote the paper in a logical manner. However, I found that in the methodology section, the authors have still chance to improve. Particularly, I hardly found the source of instrument or scale. How do they measure each variable and its source?
4. The author might provide one or two items of each variable(scale) they used.
5. This paper is based on very old literature. I will suggest to use recent citations, particularly from last 4 years.
Wish you all the best.
Author Response
Thank you very much for your helpful feedback. Please find our answers attached.

Round 2
Reviewer 2 Report
Thank you very much for the corrections and answers. Congratulations on the study.
Author Response
Thank you very much for your positive feedback and the appreciation of our corrections.
Reviewer 3 Report
Although the manuscript focuses on an interesting topic and includes adequate citations throughout the document, I still fail to understand the value of presenting information from 2008 in a scientific journal; particularly, since the circumstances have changed so significantly in the past 14 years. I am reluctant to recommend publication of this well written manuscript based solely on the age of the data presented. Therefore, I strongly recommend updating the data and consequently the analysis.
Author Response
Thank you very much for your constructive feedback. Please find our response attached.
